# Encouraging (Nudging) People to Increase Their Fluid Intake

**DOI:** 10.3390/nu15122702

**Published:** 2023-06-09

**Authors:** Charles Spence

**Affiliations:** Crossmodal Research Laboratory, Oxford University, Oxford OX2 6GG, UK; charles.spence@psy.ox.ac.uk

**Keywords:** thirst, hydration, multisensory, nudging, sensory marketing

## Abstract

This narrative historical review considers the various routes to nudging consumers towards drinking more, given self-reported evidence that many people are often not adequately hydrated. This review builds on the related notion of ‘visual hunger’. Interestingly, however, while many desirable foods are associated with distinctive sensory qualities (such as an appetizing smell), that may capture the consumer’s (visual) attention, it is less clear that there is an equivalent sensory attentional capture by hydration-related cues. One of the other important differences between satiety and thirst is that people tend to overconsume if they use interoceptive satiety cues to decide when to stop eating, while the evidence suggests that people typically stop drinking prior to being adequately hydrated. What is more, the increasing amount of time we spend in consistently warm indoor environments may also be exacerbating our need to drink more. A number of concrete suggestions are made concerning how people may be encouraged (or nudged) to imbibe sufficient water.

## 1. Introduction

Water is a critical macronutrient [1], and, as such, ensuring adequate hydration is crucial for maintaining our well-being (e.g., [2]), especially for those who enjoy (aerobic) exercise (particularly when undertaken in a warmer environment and when of extended duration/higher intensity). However, the evidence from self-report surveys suggests that many of us are failing to keep adequately hydrated, in part as a result of sensations of thirst typically being alleviated prior to full rehydration ([3]; though see also [4,5]). For instance, according to the results of one survey [6], most adults admit that they do not drink nearly enough water every day (see also [7]). Should such self-report data prove to be reliable, it would suggest that people may not always be performing either mentally or physically at their best [2,8,9]. This has led some to suggest that rather than drinking simply to alleviate thirst (referred to as homeostatic drinking), there may be occasions when programmed drinking is more appropriate (a kind of non-homeostatic drinking; [10]). It is here that the notion of nudging hydration, or the use of sensory marketing techniques, becomes relevant.

People typically spend 90–95% of their lives indoors (see [11]). At the same time, however, the evidence suggests that the indoor temperature (e.g., in our homes) has been rising as the decades have gone by ([12]; see also [13]). Additionally, while the consequences of spending so much of our time on reduced energy expenditure (and hence an increased likelihood of being overweight/obese) have already been highlighted [14], the potential effects of the increased likelihood of inadequate hydration have not been, as far as I am aware. Of course, the problem for those exercising in warm indoor environments may be even more pronounced.

When thinking about the factors that may encourage people to drink/hydrate, it is important to distinguish between the sensory properties of the drink itself (and its packaging, including any drinking spout) and those of the environment in which the drinker happens to find themselves. Both may play a significant role in helping to nudge people towards drinking more (to reach a state of optimal hydration). What is more, a distinction should also be drawn here between the decision an individual makes as to whether or not to drink in the first place, and the subsequent decision, having decided to drink, concerning when to stop. It would seem likely that different sensory strategies (or nudges) may be appropriate to increase the likelihood of consumption at each stage.

The aim of the present narrative historical review is, therefore, to consider which sensory strategies, or nudges (see [15,16], for reviews of the popular notion of ‘nudging’), may be used to help encourage people to hydrate (properly) either by inducing an awareness of thirst and/or the desire to imbibe a thirst-quenching drink or else by other sensory means. Note the parallel here with the notion of ‘visual hunger’ (see [17] for a review). That said, there would simply appear to have been far less peer-reviewed scientific research around nudging people to drink more fluid (to rehydrate themselves) as compared to work on sensorially (i.e., visually, auditorily, olfactorily, and/or thermally) nudging people to consume more (often unhealthy) food. It is, though, worth noting in passing that such psychological nudges do not always override people’s personal preferences when it comes to consumption. For instance, Venema and colleagues [18] reported that participants’ choice of a small, medium, or large soda was driven more strongly by their level of thirst (predicting relatively larger portion choices) and their level of health consciousness (predicting relatively smaller portion choices) than by the presence of a centre-stage nudge. This is the name given to the middle option in an array being chosen more frequently than those placed on the sides. At the same time, however, the relationship between informational/organizational nudges and sensory marketing remains somewhat unclear in the literature [19]. Both approaches can be used to modify people’s behaviour, though through somewhat different means. Furthermore, ethical issues concerning the use of nudging and/or sensory marketing are now starting to attract attention from researchers. 

### Thirst and Dry Mouth

Researchers have long been interested in the behavioural/perceptual correlates of thirst [10,20,21,22,23,24,25]. Researchers have also studied water as a specific response to the visceral drive of thirst in the human brain [26]. For instance, the latter researchers demonstrated that the consumption of water resulted in increased neural activation in cortical taste areas including the frontal operculum/anterior insula (the primate primary taste cortex) and the caudal orbitofrontal/secondary taste cortex. Dehydration results in a complex pattern of neural activity in the hypothalamus and higher cortical areas ([10,27]). Researchers have demonstrated that thirst neurons appear to anticipate the homeostatic consequences of eating and drinking in rats [28]. However, while several neural biomarkers associated with thirst have now been identified [29], the most obvious and direct way in which to ascertain people’s consumption behaviour is by measuring the volume consumed.

The experience/awareness of having a dry mouth may be one of the most salient sensory/perceptual cues that people use when deciding whether to hydrate because they are thirsty [30]. According to Poothullil [31], thirst, when induced by hypertonicity, will be associated with a dry oral sensation. Hence, anything that encourages salivation (such as showing someone biting into a sliced lemon) might be expected to alleviate the symptoms of dry mouth, and, hence, perhaps counterintuitively, potentially reduce an individual’s desire to hydrate (see [32], for a review). In terms of a drink’s sensory properties, McEwan and Colwill [33] conducted research showing that an acidic-tasting drink was associated most strongly with the notion of thirst-quenching, whilst sweetness and thickness were the attributes that were least strongly associated with the notion of thirst-quenching. It has often been suggested that there are appetizing food aromas, associated with high-fat, energy-dense foods (see [34]). It is, however, less obvious that specific aromas would necessarily be associated with the notion of ‘thirst-quenching’ or ‘hydration’ (and thus be used to encourage those who smell them to drink). Research by Bellows [35] found that wetting the mouth of dogs did not help to reduce thirst. However, given that dogs have special receptors to detect water on the tip of their tongues that humans do not have (see [36]), they might not necessarily provide a good model for thirst, and its alleviation, in humans. However, in an early study by Holmes and Gregersen [37], the ingestion of water prior to the introduction of thirst following the injection of a salt solution inhibited the decline in salivary flow while also preventing the appearance of thirst. A sensation of thirst may result from general rather than local dehydration. The intracellular dehydration caused by salt intake results in body fluid being lost from the cells, thus leading to a thirst response. At the same time, however, extracellular dehydration—where fluid is lost from the compartments outside of cells—may also be responsible for triggering thirst [38].

It might be interesting to target those oral sensations by asking people to introspect about how dry their oral cavity feels. Here, one could think, by analogy, of the early marketing of mouthwash, where a story was created around bad breath that suddenly made people much more aware of the state of their oral cavity than they had formerly been [39]. Simply reading olfactorily-redolent words such as “cinnamon” has been shown to give rise to activity in brain areas associated with olfaction [40]. It would seem legitimate to wonder what simply asking the question: “Are you thirsty?” might do to people. Would such a question automatically trigger introspection concerning an individual’s perceptual/bodily state? Note here that subliminally presenting thirst-related words biases people’s drinks choice, at least in the short term [41]. At the same time, encouraging people to drink through a straw or a straw-like spout will likely result in their drinking more, because we typically use the number of food/drink-related sensations to determine how much we have consumed [42,43,44]. Along similar lines, one might expect that those individuals who are distracted by digital media content, be it on video screens or their handheld devices, would end up drinking more ([45]; cf. [46]). Bar Salve Jorge provided one innovative solution targeted at reducing the distraction from mobile devices by serving beer in a glass that would not support itself unless one’s mobile device was placed under it [47].

## 2. Visual Contributions to Thirst

### 2.1. Are There Thirst-Quenching Colours?

Flavour expectations, and hence our predictions about how thirst-quenching a drink might be, are key to understanding consumption behaviour [48]. Certain colours of drink are more strongly associated in the mind of the consumer with the notion of thirst-quenching, sweetness, acceptability, and flavour intensity than others (e.g., in the case of fruit-punch-flavoured beverages; see also [49,50]). More generally, colour plays an important role in determining people’s food and drink choice behaviour [51]. There is even some limited evidence to suggest that physiological responses can be influenced by exposure to such augmented/virtual visual food stimuli [52]. In one study, Zellner and Durlach [50] assessed the effect of drink colour on expected and experienced refreshment for mint-, lemon-, and vanilla-flavoured drinks. Their results revealed that brown was judged as less refreshing than clear, blue, or green for the lemon and mint-flavoured drinks. Here, though, it should be noted that the association between colour and taste/flavour is likely based on semantics [53]. What this means, in practice, is that when a person sees a brown carbonated beverage, their mind may automatically be primed to think of a cola beverage, rather than the brown colour itself directly being associated with the notion of thirst-quenching. A similar process of semantic mediation is likely involved when seeing other distinctive drinks colours.

Other researchers have studied the effect of glass colour on the evaluation of a beverage’s thirst-quenching qualities [54,55]. According to Guéguen’s findings with 40 undergraduates, a soda beverage presented in an opaque blue or, to a lesser extent, green (i.e., cold-coloured) glass was considered more thirst-quenching than the same beverage when served in a warm red or yellow glass instead [54]. Meanwhile, Italian participants in a study by Risso et al. rated mineral water as tasting ‘fresher’ when presented in a white cup than when served in a red one, with the blue cup giving rise to intermediate ratings [55]. In a subsequent study, a white cup was preferred when the participants had to choose a drink of still water, whereas the blue cup was preferred when they were tasked with selecting a more carbonated drink of water [56]. One should also consider whether the ‘image mold’ of the packaging matches the associations in the mind of the target consumer with refreshment (e.g., [57,58,59]). Similarly, it is also relevant to ask whether a bottle, can, or perhaps other packaging format (or image mold) is most strongly linked with the notion of hydration.

One other point to consider here is the colour of the closure/drinking spout. This is presumably likely to be what the consumer will see just prior to drinking and yet is often a neglected feature of visual product/packaging design. Would a non-white drinking spout be advantageous in terms of cuing hydration? Here, one might consider only the recent development of water cups/bottles such as the Right Cup [60] and the Air-up drinking bottle [61,62]. Notice how these various drink solutions use a dominant colour of the cap/sides of the drinking vessel (combined with olfactory cues) in order to help cue the desired taste/flavour attributes. Although it might seem surprising, a number of studies published over recent decades have demonstrated that simply changing the colour of the packaging can very often change people’s perception of the sensory qualities of the contents.

### 2.2. Seeing That You Are Thirsty

Does thirst make you more likely to think you see water? Tales of thirsty desert travellers and oasis mirages are certainly a familiar trope. Speculatively, there might be an opportunity here to design a quick perceptual test that lets a consumer/athlete know, depending on what they see, whether or not they are thirsty. This suggestion is based on laboratory research demonstrating that thirst can modulate visual perception [63]. In particular, thirsty participants were more likely to see transparency (considered to be a distinctive visual feature of water) in an ambiguous image than those participants who were not thirsty (M = 58% vs. 47%). The 37 participants in the thirsty group ate a bag of salty chips immediately before the experiment (1.25 oz or 35 g, 190 kcal, 350 mg sodium) while an equal number of participants in the non-thirsty group drank water until not thirsty immediately before the experiment. Of course, more realistic/extreme manipulations of thirst might give rise to a more pronounced shift in perception. Such results fit with research from Aarts, Dijksterhuis, and De Vries [64] showing that thirsty participants (where thirst was induced by tasting and swallowing salty candies) appear to be more ‘perceptually ready’ to perceive and respond to drinking-related stimuli (such as words and drinking-related objects in a free-recall task). Note here also that there is older research suggesting that subliminal priming of branded drink choice works specifically when people are in an ‘active thirst state’ [41,65], though the effects of subliminal priming on drink choice in the thirst state may be too short-lasting to have much of an impact in a real-world setting [66].

## 3. Environmental Sensory Nudging/Marketing

Many of our behaviours are externally (or exogenously) triggered, based on environmental cues such as, for example, the smell of freshly ground coffee making you more likely to order a coffee when passing through the train station [45,67]. Even basic lighting parameters, such as lighting level (i.e., bright or dim) and hue can influence people’s choices concerning what they choose to eat [68], as well as what a drink tastes like [69], and hence potentially how much they consume. Over the last decade or so, researchers have conducted a number of studies involving immersive digital environments, and/or using virtual reality (VR) in order to assess whether and, if so, how our choice of food and drink and our subsequent perception of the taste and/or enjoyment of the consumption experience may be modified by environmental cues (e.g., see [70,71,72,73,74]).

In one representative study that is particularly relevant to the themes of the present review, Sester et al. [73] had 120 people in an immersive bar setting choose a bottle of beer (from five options) under one of two ambient conditions. One room had been designed to convey the notion of warmth/heat by showing flames (red video and ‘far west’ music), and using warm-coloured wood furniture; the other environment was much cooler, with cold blue furnishings (blue video and ‘cold’ electro music). The results demonstrated that their participants’ choice of beer (and their judgment of the appropriateness of the bar with their idea of drinking beer) was slightly, but significantly biased, by the environmental atmospheric cues (see [75], for a review of atmospheric effects on food/beverage perception). In particular, Kriek beer was chosen significantly more frequently in the cold environment than in the warm one. (There is also some limited research emerging concerning the role of ambient (indoor) temperature in biasing people’s food choices [76].) One might wonder here whether desert scenes, etc. might cue people to drink more water.

### Sensory Entrainment of Drinking to the Musical Beat

The sensory marketing literature on atmospherics has repeatedly demonstrated that people drink up to 30% more when loud fast music is played [77,78,79,80]. While the majority of such research has thus far been conducted in the context of the science laboratory, or bar/restaurant (e.g., in a very different context from the post-exercise need state imagined here), the prediction would nevertheless still have to be that similar entrainment to the musical beat would also occur when thirsty people drink (though see also [81,82]). Note that such sonic manipulations are likely to influence how much/rapidly people drink rather than the decision of whether or not to drink in the first place. Note that there have also been attempts to prime coldness through specific musical manipulations [83].

## 4. Temperature and Refreshment

The concept of ‘refreshment’ does not mean the same thing to everyone [84,85]. Indeed, both individual differences and cultural factors in the consumption of beverages likely play a role here (see [48,86,87,88]). The notion of thirst is often linked to refreshment. Indeed, part of what makes a drink ‘refreshing’ is that which is cold, with people typically associating cooler temperatures with more refreshing/thirst-quenching drinks (e.g., see Figure 10.3 in [48]; cf. [89]). The experience of coldness on the tongue appears to be an innately rewarding signal for many animals [90]. Furthermore, Brunstron and Macrae [91] highlighted the link between the temperature of a drink, the volume consumed, and the experience of a dry mouth, showing that a lower-temperature drink has a more pronounced influence on thirst reduction because of its differential effects on the post-ingestive state of the mouth.

Following this line of thought, one might consider whether making sounds of drinks being poured extra-cold might make them appear more refreshing, and hence more thirst-quenching (see also [86]). The research shows how sounds can be made to sound colder than cold—extra-chilled, in other words [92,93,94]. A drink that is seen being poured in an advert can be made to sound cooler, and hence presumably more thirst-quenching, simply by caricaturing actual pouring sounds to make a drink sound colder than cold (e.g., see ‘The sound of temperature’ for sonic examples: [95]). Another sonic direction to pursue might build off the Coca-Cola adverts from Brazil from a few years ago, where the focus was very much on the sound of the drink being poured and ice crackling in glass [96]. Alternatively, however, one might go in the opposite direction. Consider here only the ‘Coca-Cola: Try not to hear’ campaign by David [97], where the images we normally associate with a drink were presented in the absence of any sound. Do such sounds, which are presumably associated with a thirst-quenching, freshly poured cold drink, work to prime the desire to hydrate (and thus maintain an optimal state of hydration)?

The coolness of a drink can also be conveyed visually. Consider, for example, the sight of condensation on the side of a bottle, can, or glass. A few years ago, Heineken used thermochromic paint on the outside of their beer cans, to give the impression of beads of condensation on the outside of the can when the drink was a suitably low temperature to drink [58]. Meanwhile, adding carbonation, really or virtually, can also help to cue refreshment in the mind of the consumer (e.g., [84,85,98]; see also [99], for ways to enhance the sound of carbonation and the perceptual consequences of doing so; [83]). Guinard et al. [86] reported a study in which carbonation and bubble density were found to be positive determinants of the ‘thirst-quenching’ character of beers amongst 18 beers judged by 10 experts. Negative determinants of ‘thirst-quenching’ included foam, overall aroma and flavour and colour, amongst other sensory attributes.

## 5. The Social Facilitation of Drinking

It would be interesting to know whether simply showing images of people who are thirsty, or who are obviously engaging in thirst-inducing exercise/activity, would lead viewers to experience more thirst/a greater need to hydrate. Even something as simple as mounting a large mirror in front of a drink station might nudge people to drink more (cf. [100,101]). Potentially relevant here, research on the auditory mirror neuron system suggests that our motor system may imitate heard actions, such as those associated with the consumption of food ([102]; cf. [103], on the beneficial effect of lip-smacking sounds on food enjoyment). As such, hearing the sounds associated with other people drinking/rehydrating might be expected to elicit mirror system activity. If so, it is natural to extrapolate to suggest that showing other people drinking might also nudge healthy hydration.

## 6. Embodied Cognition and Simulating the Act of Drinking

According to the literature on ‘embodied cognition’ [104], consumers simulate the act of eating (and presumably also drinking) what they see, even if it is only displayed on a screen or billboard (see [17,68], for reviews). According to the literature on ‘visual hunger’, drinks should be shown close-up (i.e., as if in the viewer’s own personal space) and from a first-person perspective. Note that this strategy has been successfully used in the case of fast-food adverts in recent years (see [68], for a review), as in the advertising from Carl’s Junior showing an attention-capturing food-in-motion moment [105]. Such suggestions are designed to make it easier to simulate the act of consumption. Visual communications should therefore avoid giving the impression that the portrayed drink is someone else’s. These recommendations are all based on research showing that we like food that little bit more if the spoon, for a bowl of soup, is shown in the viewer’s dominant right hand [106]. Presenting food in motion has also been shown to attract the attention of consumers [36], and hence the sight and sound of refreshing drinks being poured will likely also be beneficial.

## 7. Recommendations

In conclusion, there are a number of directions that may work to help sensorially nudge people to hydrate more, given the repeated observation from self-report surveys that people do not think that they hydrate adequately. These suggestions target everything from the multisensory design of drinks through to the packaging in which they happen to be presented [107], and from the way they are displayed in digital/print advertising through to the context in which they are consumed [108], relating to both the atmosphere and any social facilitation of drinking behaviour. Individually, many of the suggestions would appear incremental in terms of their likely effect on healthy hydration. It remains an open question as to whether the impact of the various cues can potentially be combined to deliver a stronger nudge, or multisensory intervention (this a psychological approach), to encourage adequate hydration by priming drinking-related behaviours. One of the unique considerations to have emerged from this review is to highlight the potential impact of increasing indoor temperatures on hydration. One of the other important considerations concerns the question of whether water may itself represent an independent taste modality [109,110].

At the same time, it should also be noted how much of the research reviewed here was conducted on WEIRD participants [111], that is Western, Educated, Industrialized, Rich, and Democratic individuals which may not necessarily be representative of society as a whole. This concern becomes all the more salient when it is realized how much of the traditional research was conducted on students studying psychology [112]. One other concern looking back on the early research is how many of the studies would appear underpowered by contemporary standards (and adequate sample sizes were rarely justified). Addressing such concerns is contemporary research, showing how sonic seasoning can be used to modify the taste of an isotonic drink [113].

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
