# Peer review of "Encouraging (Nudging) People to Increase Their Fluid Intake"

_nutrients, 2023, doi:10.3390/nu15122702_

Round 1

Reviewer 1 Report (Previous Reviewer 3)

Dear author,

This is research that provides knowledge about healthy lifestyles and how these could be modified, for hydration purposes through water. This article is consistent with the "Hydration" section of Nutrients.

Thanks.

Author Response

Many thanks for the positive review. No specific changes requested.

Reviewer 2 Report (New Reviewer)

The article brings a very interesting topic regarding different mechanisms related to the act of drinking liquids.

The initial problem is the fact that the urban lifestyle promotes situations in which a state of mild and chronic dehydration can set in, impairing the optimal performance of physical and intellectual activities. Thus, several stimuli can be promoted to mitigate this situation, including visual, sound, environmental.

User-friendly text, excellent use of the English language and relevant subject for the readership of the scientific journal.

In order to make the text even more interesting, it would be necessary to change the initial part of the text that briefly talks about the neural and hormonal mechanisms related to thirst. I suggest that the mechanisms be described in more detail, even an illustrative figure that summarizes, in order to bring these mechanisms in a more technical way.

Author Response

Many thanks for the very positive review. As this is a psychological rather than neuroscientific review, I have chosen not to emphasize more the neural mechanisms.

Reviewer 3 Report (New Reviewer)

Dear Dr. Spence,

Thank you for submitting your manuscript to Nutrients.  

I have completed my evaluation of your great manuscript.

This document takes in account many aspects that other more physiological oriented papers did not.

I  invite you to improve the title because it now misaddresses the content of the article : 

as you know hydration is different from fluid intake .  

Encouraging (nudging) people to increase the fluid intake  or  Encouraging (nudging) people to improve the fluid intake

no comments,  writer is english from Oxford ! what else?!

Author Response

I have modified the title and header of paper as suggested

This manuscript is a resubmission of an earlier submission. The following is a list of the peer review reports and author responses from that submission.

Round 1

Reviewer 1 Report

Hydration has grown into a popular topic in nutrition research in recent years. This article's concept is both fresh and intriguing. The author goes into detail about how thirst, visual, environmental sensor nudging, temperature and refreshment might alter thirst experience and drinking behavior. It proposes to encourage drinking water through multisensory approach, which could be one of the techniques for enhancing bodily hydration in the future.

However, empirical research on the effect of multisensory strategy on healthy hydration is still in its early stages. The majority of the material included in this review is about how the above sensation can boost people's drinking behavior and also is frequently employed for beverage marketing or consumption promotion. Beverages increase water intake while simultaneously increasing sugar intake, which have been shown to be harmful to people’s health. This cannot be referred to as healthy hydration in the field of nutrition. Therefore, I think this work is not appropriate for publishing in this journal due to possible confusion/misleading. and is suggested to submit to magazines with more relevant themes.

Reviewer 2 Report

I read the article with great interest and thought it was well written and enjoyable to read. However, it appears to describe and investigate a variety of psychological rather than physiological aspects of nutrition and body composition. Furthermore, the manuscript lacks originality in my opinion, which is the primary reason for rejection. Furthermore, the recommendations at the end add nothing to what has already been said. For these reasons, I believe the article is not appropriate to be published in this journal.

Reviewer 3 Report

Dear authors,

Please review the document.

Thank you.
